# Improved HPLC Conditions to Determine Eumelanin and Pheomelanin Contents in Biological Samples Using an Ion Pair Reagent

**DOI:** 10.3390/ijms21145134

**Published:** 2020-07-20

**Authors:** Shosuke Ito, Sandra Del Bino, Tomohisa Hirobe, Kazumasa Wakamatsu

**Affiliations:** 1Department of Chemistry, Fujita Health University School of Medical Sciences, Toyoake 470-1192, Japan; 2L’Oreal Research and Innovation, 93600 Aulnay-sous-Bois, France; sdelbino@rd.loreal.com; 3Shinjuku Skin Clinic, 10F Shinjuku M-SQUARE, 3-24-1 Shinjuku, Shinjuku-ku, Tokyo 160-0022, Japan; tmhirobe@ae.auone-net.jp

**Keywords:** pyrrole-2,3,5-tricarboxylic acid (PTCA), pyrrole-2,3-dicarboxylic acid (PDCA), pyrrole-2,3,4,5-tetracarboxylic acid (PTeCA), thiazole-2,4,5-tricarboxylic acid (TTCA), thiazole-4,5-dicarboxylic acid (TDCA), total melanin (TM), 4-amino-3-hydroxyphenylalanine (4-AHP), alkaline hydrogen peroxide oxidation (AHPO), high-performance liquid chromatography (HPLC), individual typology angle (ITA)

## Abstract

Alkaline hydrogen peroxide oxidation (AHPO) of eumelanin and pheomelanin, two major classes of melanin pigments, affords pyrrole-2,3,5-tricarboxylic acid (PTCA), pyrrole-2,3-dicarboxylic acid (PDCA) and pyrrole-2,3,4,5-tetracarboxylic acid (PTeCA) from eumelanin and thiazole-2,4,5-tricarboxylic acid (TTCA) and thiazole-4,5-dicarboxylic acid (TDCA) from pheomelanin. Quantification of these five markers by HPLC provides useful information on the quantity and structural diversity of melanins in various biological samples. HPLC analysis of these markers using the original method of 0.1 M potassium phosphate buffer (pH 2.1):methanol = 99:1 (85:15 for PTeCA) on a reversed-phase column had some problems, including the short lifetime of the column and, except for the major eumelanin marker PTCA, other markers were occasionally overlapped by interfering peaks in samples containing only trace levels of these markers. These problems can be overcome by the addition of an ion pair reagent for anions, such as tetra-*n*-butylammonium bromide (1 mM), to retard the elution of di-, tri- and tetra-carboxylic acids. The methanol concentration was increased to 17% (30% for PTeCA) and the linearity, reproducibility, and recovery of the markers with this improved method is good to excellent. This improved HPLC method was compared to the original method using synthetic melanins, mouse hair, human hair, and human epidermal samples. In addition to PTCA, TTCA, a major marker for pheomelanin, showed excellent correlations between both HPLC methods. The other markers showed an attenuation of the interfering peaks with the improved method. We recommend this improved HPLC method for the quantitative analysis of melanin markers following AHPO because of its simplicity, accuracy, and reproducibility.

## 1. Introduction

Melanin pigments are widespread in nature from microorganisms to humans. Melanins have a diverse array of functions, including photoprotection, scavenging of reactive oxygen species, sequestering of toxic metals, thermoregulation, and camouflage [1]. Melanins can be classified as brown to black eumelanin and yellow to reddish pheomelanin [2,3]. These subgroups of melanin have distinct properties, not only with regard to their color, but also in their photochemical and other properties [4,5]. Another important aspect of melanin production is that most natural melanin pigments are co-polymers of eumelanin and pheomelanin [6,7,8]. To understand the biological roles of melanins, it is therefore essential to quantify the eumelanin and pheomelanin contents in various tissue samples. This was the reason why we developed a microanalytical method to quantify eumelanin and pheomelanin in tissue samples in 1985 using high-performance liquid chromatography (HPLC) [9]. The rationale for that assay was that permanganate oxidation of eumelanin gives pyrrole-2,3,5-tricarboxylic acid (PTCA) as a specific marker of eumelanin while hydroiodic acid (HI) hydrolysis gives aminohydroxyphenylalanine (AHP) isomers as a specific marker of pheomelanin.

The previous HPLC method was later modified to improve the versatility of the oxidation method [7] and the specificity of pheomelanin determination [10]. Eumelanin consists of 5,6-dihydroxyindole (DHI) and 5,6-dihydroxyindole-2-carboxylic acid (DHICA) moieties [11], while pheomelanin is derived mainly from benzothiazine and benzothiazole moieties [12]. Therefore, we developed an alkaline hydrogen peroxide oxidation (AHPO) method to quantitatively evaluate these melanin moieties. AHPO of eumelanin gives PTCA and pyrrole-2,3-dicarboxylic acid (PDCA) as degradation products arising from DHICA and DHI moieties, respectively, and thiazole-2,4,5-tricarboxylic acid (TTCA) and thiazole-4,5-dicarboxylic acid (TDCA) as degradation products from benzothiazole moieties of pheomelanin (Figure 1) [3,6]. In addition, we introduced pyrrole-2,3,4,5-tetracarboxylic acid (PTeCA) as a marker of cross-linking in eumelanin [13], especially in the identification of fossilized eumelanin [14,15,16] and in assessing the photoaging of eumelanin [5].

The HPLC conditions for HI hydrolysates to analyze the benzothiazine moiety of pheomelanin were improved to separate the AHP isomers, 4-amino-3-hydroxyphenylalanine (4-AHP) and 3-amino-4-hydroxyphenylalanine (3-AHP), using sodium octanesulfonate as an ion pair reagent (Figure 1) [6]. This method is highly specific to the benzothiazine moiety of pheomelanin due to the specificity of the electrochemical detection employed. On the other hand, the HPLC conditions used for the AHPO products in the original method, 0.1 M potassium phosphate buffer, pH 2.1:methanol = 99:1 at 45 °C, had some inherent problems. First, although PTCA, a major marker of eumelanin, was well separated from possible interfering peaks, other minor markers (TDCA, TTCA and PDCA) were occasionally overlapped by interfering peaks in biological samples containing only trace levels of those markers, which is a significant problem for TTCA, a major and important marker of pheomelanin [7,8]. Second, column performance (such as peak height and retention time) varied from one lot to another, especially for TTCA (peak height 4833 ± 660 arbitrary units (AU) and retention time 17.22 ± 1.50 min for five columns) and the column lifetime was rather short (less than three months) due to shortening of the retention time of the markers (as described below).

All markers, PDCA, TDCA, PTCA, TTCA and PTeCA, contain a vicinal dicarboxylic acid group, which makes them strongly acidic among carboxylic acids with low pKa values. By taking advantage of this acidity, di-, tri- and tetra-carboxylic acids can be separated well using an ion pair reagent, tetra-*n*-butylammonium bromide (TBA^+^Br^−^) [17]. Nuta et al. and Matsunaka et al. have reported the HPLC conditions using another ion pair reagent, tetra *n*-propylammonium bromide [18,19]. We describe here the efficacy of TBA^+^Br^−^ to analyze the five degradation markers in AHPO products of eumelanin and pheomelanin.

## 2. Results

### 2.1. Evaluation of the Improved HPLC Conditions

We evaluated the efficacy of TBA^+^Br^−^ for analysis of the eumelanin markers PTCA, PDCA and PTeCA, and the pheomelanin markers TTCA and TDCA in synthetic melanins and in biological samples containing eumelanin and pheomelanin. After some preliminary trials, we established 1 mM TBA^+^Br^−^ in 0.1 M potassium phosphate buffer, pH 2.1:methanol = 83:17 (*v*/*v*) (the improved method) at 40 °C as the optimal condition for the separation of PTCA, TTCA, PDCA and TDCA, which had retention times of 24.6, 29.3, 15.3 and 16.6 min, respectively. It should be pointed out that in the new method, TTCA emerges after PTCA in contrast to the original method in which TTCA emerged shortly after TDCA. This reduces the possible interference from other peaks in the improved method. Figure 2 shows HPLC chromatograms of the melanin markers and the AHPO mixtures from human black hair, human red hair and dark human epidermis. TTCA was split into two peaks in the original standard solution. The minor peak was reduced in peak height from 11% to 5% of the major peak when a solution mimicking the AHPO mixture was used to dissolve the melanin markers (see Materials and Methods). To separate PTeCA, we used 1 mM TBA^+^Br^−^ in 0.1 M potassium phosphate buffer, pH 2.1:methanol = 70:30 (*v*/*v*) at 40 °C. Under these conditions, PTeCA eluted at 31.3 min while PDCA, TDCA, TTCA and PTCA eluted at 6.4, 6.4, 6.8 and 7.2 min, respectively; in addition to PTeCA, the major melanin marker PTCA was well separated from other three markers.

Considerable improvements or modifications from the original method [7] were made in this new method. Firstly, based on the study of Rioux et al. [20], the wavelength used to detect the markers was changed from 269 nm to 272 nm, by which PTCA, PDCA and TTCA gave higher peaks while TDCA gave a lower peak. Secondly, following the AHPO, the oxidation mixtures were acidified with 6 M H_3_PO_4_ (150 μL) instead of 6 M HCl (140 μL). This is because we noticed some fluctuation in the retention times of the four markers following acidification with 6 M HCl since separation of the markers with the improved method is very sensitive to pH. The use of 6 M H_3_PO_4_ allowed the pH of each reaction mixture used for injection to be constant at pH 3.0, rendering the retention times constant. The linearity of the peak heights of the five markers with the improved method was excellent from 1 ng to 80 ng (2 ng to 160 ng for PTeCA) per injection when 80 μL of each marker were injected (Table 1). Calibration curves were linear for PTCA, PDCA, TTCA, and TDCA between 12.5 and 1000 ng/mL and PTeCA between 25 and 2000 ng/mL with R^2^ ≥ 0.9997. The limits of detection (LOD) for PTCA, PDCA, TTCA, and TDCA ranged from 5 to 12 ng/mL, except for PTeCA (45.7 ng/mL). The limits of quantification (LOQ) for PTCA, PDCA, TTCA, and TDCA ranged from 16 to 37 ng/mL, except for PTeCA (139 ng/mL).

The reproducibility of the improved method was good to excellent, with a coefficient of variation (CV) < 5% for most of the markers from human black and red hair (Table 2). This level of reproducibility is ascribed to the direct injection of the AHPO mixtures into the HPLC column without any pretreatment except for the addition of the reducing agent, Na_2_SO_3_, followed by the acidification with 6 M H_3_PO_4_. High CV for PTeCA in red hair and TDCA in black hair may be due to the lower level of PTeCA and TDCA and the lower peak height of PTeCA and TDCA standards, respectively.

The recovery of the added standards was examined next. The AHPO mixture from human red hair was spiked with 50 ng/mL of each of the five standards and recoveries were evaluated. The recovery was quantitative with good reproducibility (Table 3).

The column performance was compared for the retention time of PTCA. A column used with the improved method showed an increase of retention time by 0.011 min (0.05% of retention time) per sample from 23.61 to 23.71 min during the analysis of 10 samples, while a column used with the original method showed a retention time that was decreased by 0.043 min (0.15% of retention time) per sample from 27.77 to 27.38 min. This rapid decrease in retention time with the original method rendered the lifetime of the columns rather short.

### 2.2. Correlation of the Improved HPLC Conditions with the Original HPLC Conditions

First, we compared the improved HPLC method with the original method using synthetic melanins. These model melanins (dopa+cys melanins) were prepared by the oxidation of dopa in the presence of varying molar ratios of cysteine (1:0, 1:0.5, 1:1) by mushroom tyrosinase [3]. We also prepared another set of model melanins (DHI+DHICA melanins) from varying molar ratios of DHI and DHICA (1:0, 1:1, 0:1). These melanin preparations were heated as a powder at 100 °C for eight days to mimic the structural modification process of hair melanins [7,21]. This treatment leads to decarboxylation and cross-linking in eumelanin [13] and converts a majority of the benzothiazine moieties in pheomelanin to benzothiazole moieties [7]. The 12 synthetic melanins were subjected to AHPO followed by HPLC and the results of the chemical characterization are presented in Appendix A.

As summarized in Figure 3, under the improved HPLC conditions, all five markers showed good to excellent correlations with slopes close to 1.0 to the original HPLC conditions. There were some outliers in the correlation graphs of PDCA due to interfering peaks in the original method.

Next, we examined five kinds of mouse hair samples (*n* = 10) of various genetic origins (black, brown, slaty, pink-eyed dilution, and recessive yellow) [22]. Mouse hair samples of various colors were examined for correlations (Appendix A). Figure 4 shows good to excellent correlations between both methods with slopes close to 1.0.

Then, we examined human hair samples using the original and the improved HPLC methods. Human hair samples (*n* = 12) of various colors (black, dark brown, medium brown, light brown, blond and red) were examined for correlations (Appendix A). Figure 5 shows good to excellent correlations with slopes close to 1.0 for both methods. It should be noted that PTeCA values with the original HPLC method showed a background value of 9 ng/mg due to an interfering peak. TTCA values also had a background of 5 ng/mg.

Finally, human epidermal samples (*n* = 18) with varying degrees of constitutive pigmentation, ranging from very light to dark, were examined for correlations. These epidermal samples are identical to those used in our previous study (*n* = 35) [8] but were selected for this study. Figure 6 shows good to excellent correlations with slopes close to 1.0 with both HPLC methods for PTCA, PTeCA, and TTCA. PDCA values were not compared because a peak interfered with PDCA in the original HPLC method. PTeCA and TTCA values in the original method had background values of 2.2 ng/mg and 1.4 ng/mg, respectively. Also, there was a small interfering peak appearing close to TDCA in the improved HPLC method. The results of detailed chemical analyses are presented in Appendix A and Appendix A, which gave results closely resembling those in the previous study [8]. Figure 2D illustrates a typical HPLC chromatograph for dark epidermis.

Correlations of PTCA, TTCA and 4-AHP to Individual Typology Angle (ITA), a measure of objective color assessment [23,24], are presented in Appendix A. Except for 4-AHP (R^2^ = 0.355), PTCA (R^2^ = 0.920) and TTCA (R^2^ = 0.913) values showed good correlations with ITA values. Total melanin (TM) values by spectrophotometry [25] and HPLC as a sum of eumelanin, benzothiazole- pheomelanin and benzothiazine-pheomelanin showed excellent (R^2^ = 0.922 for spectrophotometry; 0.927 for HPLC) correlations with ITA values and the correlations between TM values obtained by spectrophotometry and by HPLC were also excellent (R^2^ = 0.983) (Figure 7). We obtained similar results in a previous study [8], but the correlation coefficients are much better with the improved HPLC method, indicating the reliability of the improved method in terms of accuracy and reproducibility.

## 3. Discussion

Nuta et al. [18], examining the effects of ion pair reagents for anionic compounds, showed that the PTCA peak is retarded several-fold more by TBA^+^Br^−^ than by tetra *n*-propylammonium bromide and PTCA is retarded more at lower pH. Based on these preliminary results, we chose to add 1 mM TBA^+^Br^−^ to the original HPLC condition of 0.1 M potassium phosphate buffer, pH 2.1 [7]. An improved method for separating specific melanin degradation markers (PTCA, PDCA, PTeCA, TTCA and TDCA) has been developed based on the use of TBA^+^Br^−^, an ion pair reagent, to retain anionic compounds on reversed-phase HPLC columns [17]. As shown in Figure 3, Figure 4, Figure 5 and Figure 6, PTCA values were found to be reliable both with the original and the improved HPLC methods for synthetic melanins, mouse hair samples, human hair samples and human epidermal samples. The TTCA values obtained were also found to be similar between both methods. In addition, not only PTCA values, but also TTCA values, showed high reproducibility (Table 2). This is a significant improvement of the newly developed HPLC method because PTCA and TTCA are major markers of eumelanin and pheomelanin, respectively [7,8]. The PDCA, PTeCA and TDCA values obtained are similar between both methods, although we occasionally observed artificially higher values (outliers) with the original HPLC method in synthetic melanin, mouse hair, human hair, and epidermal samples (Figure 3, Figure 4, Figure 5 and Figure 6). In HPLC chromatograms of the improved method, the TTCA peak appears at the end of the four markers, reducing the possibility of interference. The possibility of overlap by peaks interfering with PDCA and TDCA had been reduced. The lifetime of the column becomes longer due to the stable retention time of the markers. There is a possibility that the PDCA peak might be overlapped by an interfering peak. However, this possibility could be confirmed by changing column temperature by 5 to 10 °C so that the PDCA peak is separated from the interfering peak.

Using the application software by chemicalize.org (https://chemicalize.com/welcome), the strongest acidic pKa of PDCA, TDCA, PTCA, TTCA and PTeCA were 3.64, 3.22, 3.20, 2.70, and 1.11, respectively (Appendix A). The microspecies distribution (%) of mono anionic species at pH 2.1 for PDCA, TDCA, PTCA, TTCA and PTeCA, were calculated to be 2.8%, 7.1%, 7.4%, 20.9%, and 91.1%, respectively. This distribution of mono anionic species corresponds exactly to the order of elution in HPLC: PDCA < TDCA < PTCA < TTCA < PTeCA.

In recent years, interest has increased in accurately analyzing melanin markers in diverse biological samples. Affenzeller et al. [26] introduced an HPLC-UV-MS method to identify and quantify melanin markers following solid-phase extraction (reversed-phase) with >70% recoveries. Rioux et al. [20] developed an HPLC-diode array detection method for melanin markers following solid-phase extraction (weak anion exchange) with >90% recoveries. Another method introduced by Lerche et al. [27] used HPLC-MS/MS following solid-phase extraction (weak anion exchange) with the use of an internal standard, 4-methylpyrrole-2,3,5-tricarboxylic acid. The LOQ of four markers estimated by the methods reported by Rioux et al. [20] and Affenzeller et al. [26] were from 40 to 60 ng/mL and from 80 to 330 ng/mL, respectively. The LOQ reported by Lerche et al. was estimated to be 1 ng/mL for those markers [27].

These newly developed methods all employ solid-phase extraction to remove possible interfering compounds, which is a significant improvement over our original HPLC method [7]. However, pre-purification is an extra step in quantifying melanin markers which causes some variations in recovery and takes extra time. Compared to solid-phase extraction, our direct injection into the HPLC gave not only a complete recovery as there was no need for the extraction, but also a greater sensitivity and a greater reproducibility. Furthermore, our HPLC method, either the original or the improved version, avoids that extra step and adapts to a direct injection (up to 80 μL of the AHPO mixture) into the HPLC. With the improved HPLC method, the linearity of the five melanin markers is excellent and the recoveries of the added standards are quantitative. The use of an ion pair reagent such as TBA^+^Br^−^ removes most interfering compounds because the improved HPLC method utilizes 17% (*v*/*v*) methanol in place of the 1% (*v*/*v*) methanol used in the original HPLC method.

## 4. Materials and Methods

### 4.1. Materials

The melanin markers, PTCA, PDCA, TTCA and TDCA were prepared in our laboratory as described in d’Ischia et al. [3]. PTeCA was a kind gift from John D. Simon, Department of Chemistry, Duke University, USA [14]. Tetra *n*-butylammonium bromide (TBA^+^Br^−^) was purchased from Tokyo Chemical Industry Co. (Tokyo, Japan). HPLC grade methanol was from Fujifilm Wako Pure Chemical Corp. (Osaka, Japan). Other chemicals were of the highest purity commercially available.

Standard solutions containing PTCA, PDCA, TTCA and TDCA at concentrations of 1 μg/mL were prepared by diluting a 100 μg/mL aqueous stock solution of each melanin marker in medium mimicking the AHPO mixture, and were prepared by mixing 2.5 mL water, 7.5 mL 1 M K_2_CO_3_, 1.0 mL 10% Na_2_SO_3_, and 3.0 mL 6 M H_3_PO_4_ (pH 2.96). Similarly, standard solutions containing PTeCA, PTCA and PDCA at concentrations of 1 μg/mL (2 μg/mL for PTeCA) were prepared for analyzing those compounds.

### 4.2. HPLC Conditions

The HPLC system consisted of a JASCO 880-PU pump (JASCO Co., Tokyo, Japan), a C18 column (Capcell Pak MG; 4.6 × 250 mm; 5 µm particle size, Osaka Soda, Osaka, Japan) and a JASCO UV detector (JASCO Co., Tokyo, Japan) at 272 nm for PTCA, PDCA, TTCA and TDCA and at 269 nm for PTeCA (and PTCA and PDCA). The mobile phase was 0.1 M potassium phosphate buffer, pH 2.1, containing 1 mM TBA^+^Br^−^:methanol at 83:17 (*v*/*v*) for PTCA, PDCA, TTCA and TDCA and at 70:30 (v/v) for PTeCA (and PTCA and PDCA). Analyses were performed at 40 °C and at a flow rate of 0.7 mL/min. Hydroiodic acid (HI) hydrolysis was performed as described in Wakamatsu et al. [11] and soluene-350 solubilization as described in Ozeki et al. [25].

### 4.3. Melanin Analyses

The AHPO was performed as described in Ito et al. [7], except that the reaction was terminated by adding 150 µL 6 M H_3_PO_4_ instead of 140 µL 6 M HCl after adding 50 µL 10% Na_2_SO_3_. The samples used included 12 synthetic melanins (see Figure 3 legend), 5 congenic C57BL/10JHir (B10) mouse hairs (black, *a*/*a*; brown, *b*/*b*; recessive yellow, *e*/*e*; pink-eyed dilution, *p*/*p*; slaty, *slt*/*slt*) [22], 12 human hair samples of various colors (2 each of black, dark brown, brown, light brown, blond and red) obtained from ColoRight Co. (Tel Aviv, Israel), and 18 human epidermal samples with varying degrees of pigmentation (3 each of very light, light, intermediate, tan, brown and dark). A detailed description of these epidermal samples is presented in Del Bino et al. [8].

Quantities of samples subjected to AHPO were 0.2 mg, 1.0 mg, 1.0 mg and 1.0 mg for synthetic melanin, mouse hair, human hair and human epidermis, respectively. The samples were homogenized in water using a Ten-Broeck glass homogenizer.

### 4.4. Statistical Analyses

Linear regression analysis was employed to determine the coefficient of determination R^2^ and *p*-value for the slope coefficient with Microsoft Excel and JMP 10 software (SAS Institute Inc., Cary, NC, USA), respectively.

## 5. Conclusions

PTCA analysis is reliable with the original HPLC method as well as the improved HPLC method. However, for the analysis of the other melanin markers, particularly TTCA and PTeCA, the improved HPLC method is significantly better for specificity. Thus, we recommend this improved HPLC method for the quantitative analysis of melanin markers following AHPO because of its simplicity, specificity and reproducibility.

## Figures and Tables

**Figure 1 ijms-21-05134-f001:**
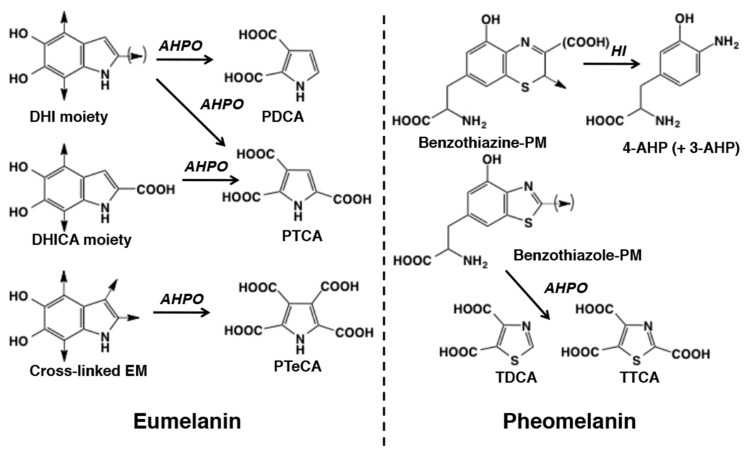
Alkaline hydrogen peroxide oxidation (AHPO) and hydroiodic acid (HI) hydrolysis of eumelanin and pheomelanin. 5,6-dihydroxyindole (DHI), 5,6-dihydroxyindole-2-carboxylic acid (DHICA), pyrrole-2,3-dicarboxylic acid (PDCA), pyrrole-2,3,5-tricarboxylic acid (PTCA), pyrrole-2,3,4,5-tetracarboxylic acid (PTeCA), thiazole-4,5-dicarboxylic acid (TDCA), thiazole-2,4,5-tricarboxylic acid (TTCA), 4-amino-3-hydroxyphenylalanine (4-AHP), 3-amino-4-hydroxyphenylalanine (3-AHP).

**Figure 2 ijms-21-05134-f002:**
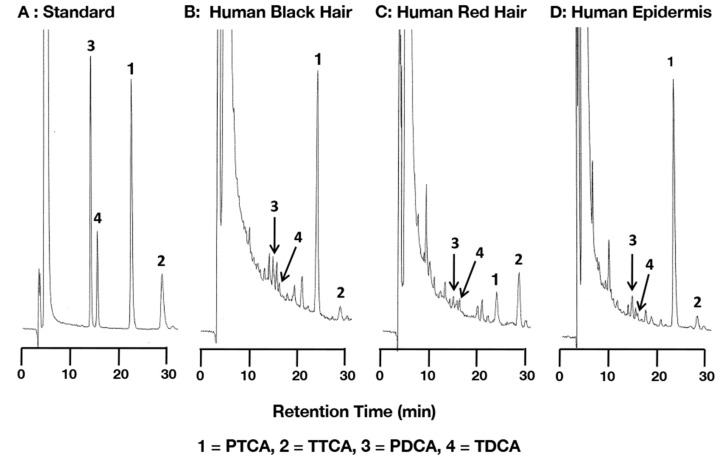
High-performance liquid chromatography (HPLC) chromatograms of standard melanin markers (**A**), and AHPO mixtures from human black hair (**B**), from human red hair (**C**) and from dark human epidermis (**D**).

**Figure 3 ijms-21-05134-f003:**
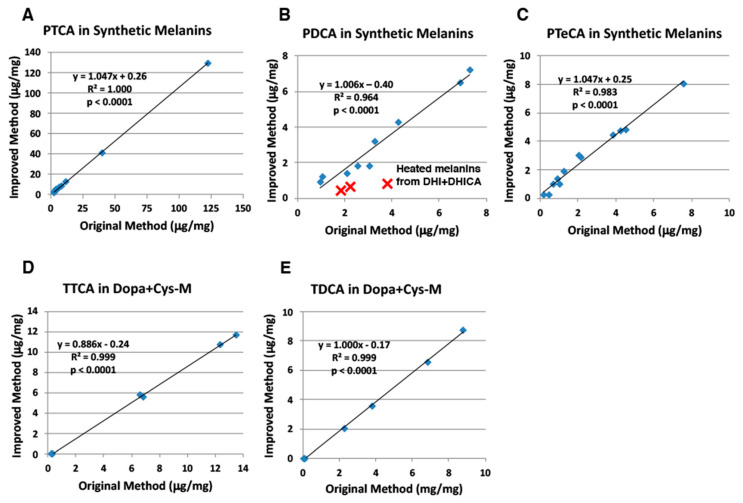
Correlations between the original and improved HPLC methods for analysis of synthetic melanins. (**A**) PTCA, (**B**) PDCA, (**C**) PTeCA, (**D**) TTCA, and (**E**) TDCA. Synthetic melanins were prepared by tyrosinase oxidation of mixtures of dopa and cysteine (cys) at ratios of 1:0, 1:0.5 and 1:1 and of mixtures of DHI and DHICA at ratios of 1:0, 1:1 and 0:1, and were then heated at 100 ℃ for 8 days (aged melanins) [21]. The AHPO mixtures were analyzed both by the original HPLC method [7] and by the improved HPLC method (this study). Outliers are indicated by red x. Units for *x*- and *y*-axis are μg/mg. Dopa+Cysteine melanins (Dopa+Cys-M).

**Figure 4 ijms-21-05134-f004:**
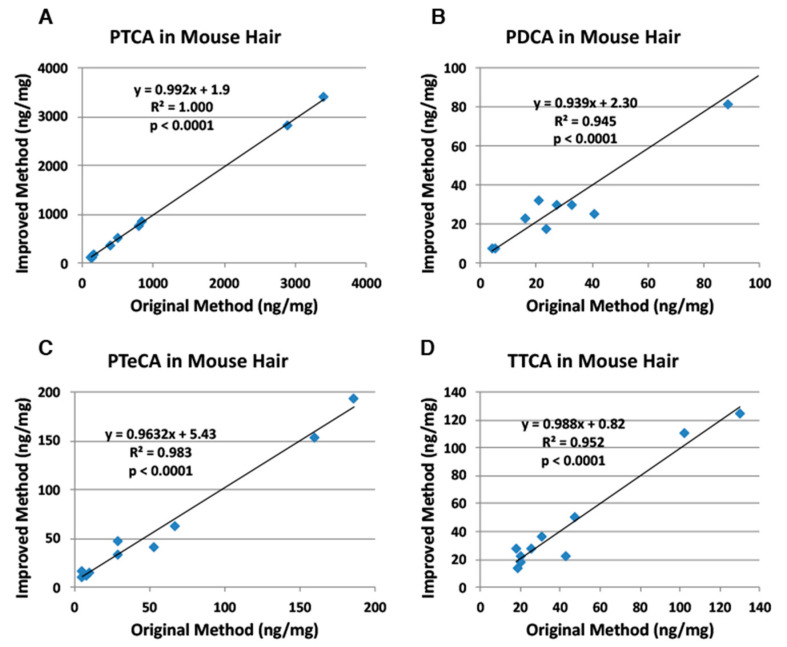
Correlations between the original and improved HPLC methods for AHPO of mouse hair samples. (**A**) PTCA, (**B**) PDCA, (**C**) PTeCA, and (**D**) TTCA. The AHPO mixtures were analyzed both by the original HPLC method [7] and by the improved HPLC method (this study). Units for *x*- and *y*-axis are ng/mg.

**Figure 5 ijms-21-05134-f005:**
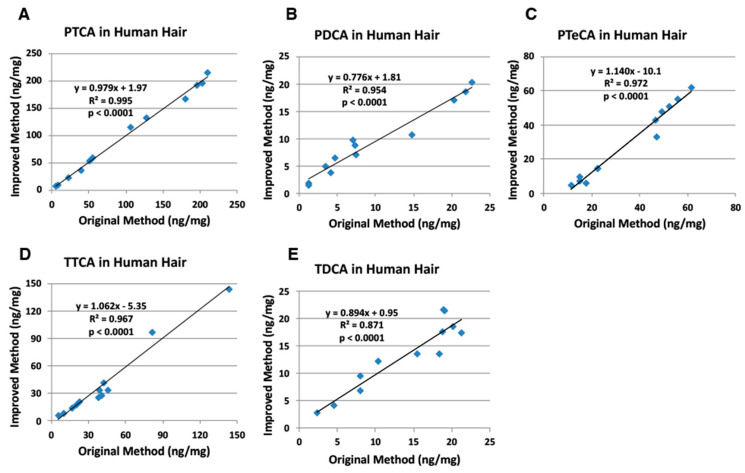
Correlations between the original and improved HPLC methods for AHPO of human hair samples. (**A**) PTCA, (**B**) PDCA, (**C**) PTeCA, (**D**) TTCA, and (**E**) TDCA. The AHPO mixtures were analyzed both by the original HPLC method [7] and by the improved HPLC method (this study). Units for *x*- and *y*-axis are ng/mg.

**Figure 6 ijms-21-05134-f006:**
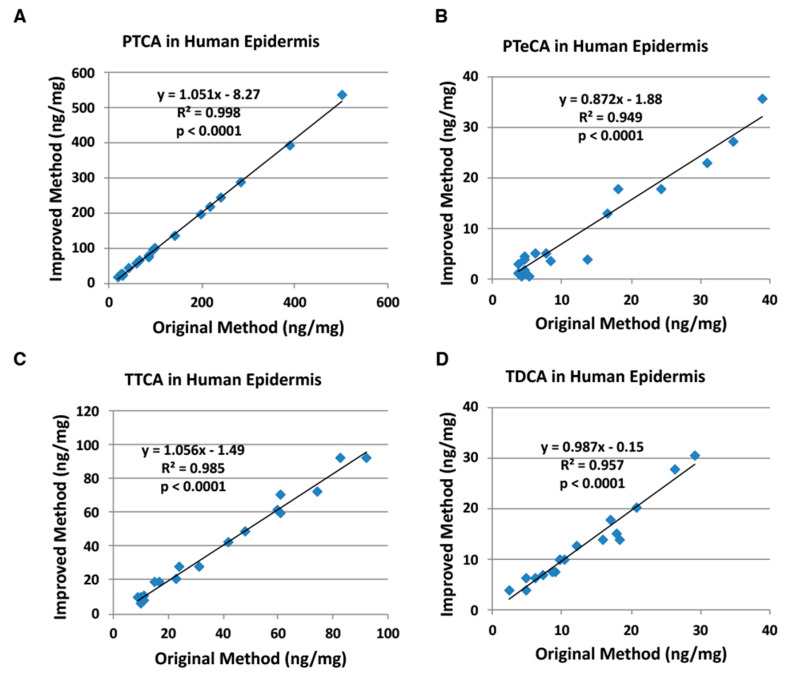
Correlations between the original and improved HPLC methods for human epidermal samples. (**A**) PTCA, (**B**) PTeCA, (**C**) TTCA, and (**D**) TDCA. The AHPO mixtures were analyzed both by the original HPLC method [7] and by the improved HPLC method (this study). Units for *x*- and *y*-axis are ng/mg.

**Figure 7 ijms-21-05134-f007:**
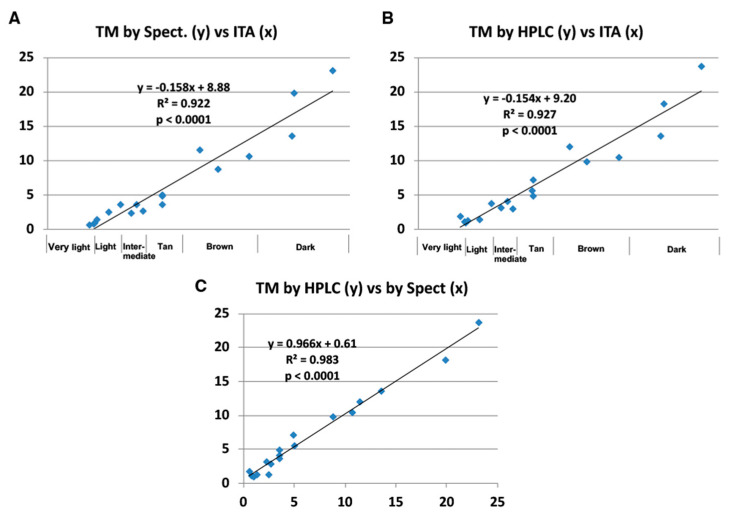
Correlation between Individual Typology Angle (ITA) and melanin markers in human epidermal samples. (**A**) Total melanin (TM) by spectrophotometry calculated by multiplying the A500 value (1/mg) by 135 µg [8] vs. ITA. (**B**) TM by HPLC as the sum of eumelanin, benzothiazole-pheomelanin and benzothiazine-pheomelanin, calculated by multiplying PTCA, TTCA and 4-AHP values (ng/mg) by 0.038, 0.034 and 0.009, respectively [8] vs. ITA. (**C**) Correlation of TM values between spectrophotometry and HPLC. Units for the *y*-axis are µg/mg. Actual ITA values (greater and positive values for paler skin and smaller and negative values for darker skin) [8,23,24] are converted to positive values by values being reversed and the *y*-axis being moved to the left end [8]. Correlation formulae are based on actual values of ITA.

**Table 1 ijms-21-05134-t001:** Validation of the improved method with respect to linearity of calibration, LOD and LOQ.

Marker	Linearity Range (ng/mL)	Regression Equation	Correlation (R^2^)	LOD (ng/mL)	LOQ (ng/mL)
PTCA	12.5–1000	y = 15.29x + 0.9	1.0000	11.1	33.7
PDCA	12.5–1000	y = 15.76x + 8.1	1.0000	12.3	37.2
TTCA	12.5–1000	y = 4.06x + 13.1	1.0000	5.17	15.7
TDCA	12.5–1000	y = 6.28x + 8.7	1.0000	11.5	34.7
PTeCA	25–2000	y = 6.85x − 5.2	0.9997	45.7	139

Note 1: 80 µL of standard solutions containing 12.5 to 1000 ng/mL of each marker (12.5, 25, 50, 100, 250, 500, and 1000 ng/mL) except for PTeCA (25 to 2000 ng/mL) were injected. Values were calculated for means of duplicate analyses. Note 2: LOD (limits of detection) set at 3.3:1 and LOQ (limits of quantification) at a 10:1 signal-to-noise ratio.

**Table 2 ijms-21-05134-t002:** Reproducibility of melanin markers with the improved HPLC method (*n* = 5).

Hair	PTCA	PDCA	PTeCA	TTCA	TDCA
Black	205 ± 4.4 (2.1%)	19.2 ± 0.60 (3.2%)	82.0 ± 3.6 (4.4%)	43.4 ± 0.94 (2.2%)	19.4 ± 1.9 (9.7%)
Red	23.4 ± 1.2 (5.2%)	10.7 ± 0.52 (4.8%)	12.9 ± 1.6 (12.4%)	169 ± 6.7 (4.0%)	26.7 ± 0.86 (3.2%)

Values are means ± SD in ng/mg; coefficient of variations (CVs) are shown in parentheses.

**Table 3 ijms-21-05134-t003:** Recovery of melanin markers with the improved method (*n* = 3).

Hair	PTCA	PDCA	PTeCA	TTCA	TDCA
**Red**	103.0 ± 2.5%	106.8 ± 1.6%	95.6 ± 7.6%	89.6 ± 4.5%	96.3 ± 4.7%

A standard solution of the five markers containing 50 ng/mL each was spiked into the AHPO mixture from human red hair, and the recovery rate of five markers was measured.

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
