# Peer review of "Improved HPLC Conditions to Determine Eumelanin and Pheomelanin Contents in Biological Samples Using an Ion Pair Reagent"

_ijms, 2020, doi:10.3390/ijms21145134_

Round 1

Reviewer 1 Report

This study aimed to use an ion-pair reagent for anions, tetra-n-butylammonium bromide to retard the elution of di-, tri- and tetra-carboxylic acids and improve the analysis of 5 markers of melanin in various biological samples by HPLC.

  1. As mentioned in the literature and this manuscript, several work reported the analysis methods of eumelanin and pheomelanin contents. Therefore, the advance of this study must state clearly.
  2. There biological samples including 2 hair samples and 1 dermal sample were applied in this study for detected the 5 markers of melanin with the ion pair reagent method. The application of this study was limited. More test samples from different parts of the body or ethnicity have to test in this study.
  3. The description in Line 155” Figure 2D illustrates a typical HPLC chromatograph for dark epidermis.”. Is the Figure mentioned in this sentence correct?

Author Response

Reviewer 1:

First of all, I would like to thank four reviewers for their valuable comments.

1) We added one person (Dr. Tomohisa Hirobe) as co-author because we performed the additional experiment by using the mouse hair samples supplied from him.

2) We added the new Figure 4 showing the correlations between both methods for AHPO of mouse hair, Supplemental Table S2 showing comparison of levels of melanin markers in mouse hair samples between both methods, and Supplemental Figure S3 showing pKa of five markers.

3) Figure 3 was amended.

4) We added the new Table 1 showing validation of the improved method with respect to linearity of calibration, LOD, and LOQ.

This study aimed to use an ion-pair reagent for anions, tetra-n-butylammonium bromide to retard the elution of di-, tri- and tetra-carboxylic acids and improve the analysis of 5 markers of melanin in various biological samples by HPLC.

  • As mentioned in the literature and this manuscript, several work reported the analysis methods of eumelanin and pheomelanin contents. Therefore, the advance of this study must state clearly.

We added the following sentence:

Line 224: Compared to the solid phase extraction, our direct injection to the HPLC gave not only a complete recovery because of no need of the extraction but also a greater sensitivity and a greater reproducibility.

  • Three biological samples including 2 hair samples and 1 dermal sample were applied in this study for detected the 5 markers of melanin with the ion pair reagent method. The application of this study was limited. More test samples from different parts of the body or ethnicity have to test in this study.

We added the following sentence:

Line 152: Next, we examined five kinds of mouse hair samples (n = 10) of various genetic origins, black, brown, slaty, pink-eyed dilution, and recessive yellow [22]. Mouse hair samples of various colors were examined for correlations (Table S2). Figure 4 shows good to excellent correlations between both methods with slopes close to 1.0.

  • The description in Line 155” Figure 2D illustrates a typical HPLC chromatograph for dark epidermis.”. Is the Figure mentioned in this sentence correct?

Yes, this figure is correct.

Reviewer 2 Report

This report focuses on an improved method for quantifying melanin-related pigments in biological and synthetic samples. This report underscores the prior methods, and their drawbacks, for eumelanin and phaeomelanin concentration. The improved method, presented here, uses HPLC for quantitative analysis following treatment with AHPO to quantify a number of markers of each type of melanin. An improved method of quantification could be useful broadly, from genetic to genomic to clinical and cosmetic applications. Generally, the report is well written, however I have some comments the authors should consider below in their revision.

How do the authors explain the higher coefficient of variation for PTeCA production in red hair? Is this due to the fact that red hair is largely composed of phaeomelanin?

Similarly, is the higher CV for TDCA in black hair due to the fact that this marker is associated with phaeomelanin, which would be anticipated to be in shorter supply in black hair?

I was impressed by the experimental approach using different forms of hair color as a test to quantify eumelanin and phaeomelanin from natural substrate. Can the authors comment on the extent to which phaeomelanin is present in dark hair, and conversely, the extent to which eumelanin is present in red/light colored hair?

How does the use of synthetic pigments resolve discrepancies or CV present in the biological samples?

Author Response

First of all, I would like to thank four reviewers for their valuable comments.

1) We added one person (Dr. Tomohisa Hirobe) as co-author because we performed the additional experiment by using the mouse hair samples supplied from him.

2) We added the new Figure 4 showing the correlations between both methods for AHPO of mouse hair, Supplemental Table S2 showing comparison of levels of melanin markers in mouse hair samples between both methods, and Supplemental Figure S3 showing pKa of five markers.

3) Figure 3 was amended.

4) We added the new Table 1 showing validation of the improved method with respect to linearity of calibration, LOD, and LOQ.

Reviewer 2:

This report focuses on an improved method for quantifying melanin-related pigments in biological and synthetic samples. This report underscores the prior methods, and their drawbacks, for eumelanin and phaeomelanin concentration. The improved method, presented here, uses HPLC for quantitative analysis following treatment with AHPO to quantify a number of markers of each type of melanin. An improved method of quantification could be useful broadly, from genetic to genomic to clinical and cosmetic applications. Generally, the report is well written, however I have some comments the authors should consider below in their revision.

  • How do the authors explain the higher coefficient of variation for PTeCA production in red hair? Is this due to the fact that red hair is largely composed of phaeomelanin?
  • Similarly, is the higher CV for TDCA in black hair due to the fact that this marker is associated with phaeomelanin, which would be anticipated to be in shorter supply in black hair?

High CV may be due to the lower level of PTeCA and the lower peak height of the standard. Similarly, High CV may be also due to the lower level of TDCA and the lower peak of the standard.

Thus, I added the following sentence:

Line 125: High CV for PTeCA in red hair and TDCA in black hair may be due to the lower level of PTeCA and TDCA and the lower peak height of PTeCA and TDCA standards, respectively.

  • I was impressed by the experimental approach using different forms of hair color as a test to quantify eumelanin and phaeomelanin from natural substrate. Can the authors comment on the extent to which phaeomelanin is present in dark hair, and conversely, the extent to which eumelanin is present in red/light colored hair?

We had an opportunity to analyze a large (n = 228) set of human hair samples taken from students at the University of Arizona. (Valenzuela RK et al. Predicting phenotype from genotype: normal pigmentation. J Forensic Sci 2010; 55: 315–322.; Ito S et al. Usefulness of alkaline hydrogen peroxide oxidation to analyze eumelanin and pheomelanin in various tissue samples: application to chemical analysis of human hair melanins. Pigment Cell Melanoma Res 2011; 24: 605–613.). Black to blond hairs contained small amounts of pheomelanin at nearly constant levels (0.85–0.99 µg/mg) while eumelanin contents varied greatly depending on the intensity of colour from black, dark brown, medium brown, light brown to blond, with levels of 22.2, 14.6, 10.4, 8.7 and 4.7 µg/mg, respectively. Thus, these hairs are eumelanic, despite their great diversity in colour. The low but constant levels of pheomelanin fit well with the casing model of mixed melanogenesis. (Ito S, Wakamatsu K. Chemistry of mixed melanogenesis – pivotal roles of dopaquinone. Photochem Photobiol 2008; 84: 582–592.; Simon JD, Peles D, Wakamatsu K, Ito S. Current challenges in understanding melanogenesis: bridging chemistry, biological control, morphology, and function. Pigment Cell Mel Res 2009; 22: 563–579.) However, only red hairs contained comparable levels of eumelanin and pheomelanin at 3.8 and 4.7 µg/mg, respectively.

  • How does the use of synthetic pigments resolve discrepancies or CV present in the biological samples?

Artificially higher values by the original method (outliers) have been corrected by the improved method.

Reviewer 3 Report

In general terms, the manuscript has several conceptual and experimental deficiencies. There is no rigorous statistical analysis, very important factors such as LOD and LOQ are not determined, particularly in this system with such a marked matrix effect. Images should be improved in resolution and clarity and should be discussed in a more appropriate way. The conclusions say that they found a much better method than they had before but the results show no clear evidence of it. There is no conclusive contribution of knowledge, since the results they obtain are very similar to those they had previously, and some even show a low correlation.

Author Response

First of all, I would like to thank four reviewers for their valuable comments.

1) We added one person (Dr. Tomohisa Hirobe) as co-author because we performed the additional experiment by using the mouse hair samples supplied from him.

2) We added the new Figure 4 showing the correlations between both methods for AHPO of mouse hair, Supplemental Table S2 showing comparison of levels of melanin markers in mouse hair samples between both methods, and Supplemental Figure S3 showing pKa of five markers.

3) Figure 3 was amended.

4) We added the new Table 1 showing validation of the improved method with respect to linearity of calibration, LOD, and LOQ.

Reviewer 3:

  • In general terms, the manuscript has several conceptual and experimental deficiencies. There is no rigorous statistical analysis, very important factors such as LOD and LOQ are not determined, particularly in this system with such a marked matrix effect.

We calculated the LOD and LOQ and prepared a new Table 1 for Linearity range, Regression equation, Correlation, LOD, and LOQ, and replaced old Figure 3 with this Table 1.

We added the following sentence in the text:

Line 117: Calibration curves were linear for PTCA, PDCA, TTCA, and TDCA between 12.5 and 1000 ng/mL and PTeCA between 25 and 2000 ng/mL with R2 ≧ 0.9997. LOD for PTCA, PDCA, TTCA, and TDCA was ranged from 5 to 12 ng/mL, respectively, except for PTeCA (45.7 ng/mL). LOQ for PTCA, PDCA, TTCA, and TDCA was ranged from 16 to 37, except for PTeCA (139 ng/mL).

We added the following sentence in discussion:

Line 217: LOQ of four markers estimated by methods reported by Rioux et al. [20] and Affenzeller et al. [26] were from 40 to 60 ng/mL and from 80 to 330 ng/mL, respectively. LOQ reported by Lerche et al. was estimated to be 1 ng/mL for those markers [27].

2) Images should be improved in resolution and clarity and should be discussed in a more appropriate way.

All images of figures were improved with 400 pixel/cm.

3) The conclusions say that they found a much better method than they had before but the results show no clear evidence of it.

We added the following sentence in the text:

Line 199: In HPLC chromatogram of the improved method TTCA peak appears at the end of four markers, reducing a possibility of interference. Possibility of overlap by peaks interfering with PDCA and TDCA has been reduced. Life-time of the column becomes longer due to the stable retention time of the markers. However, this possibility could be confirmed by changing column temperature by 5 to 10ËšC so that PDCA peak is separated from the interfering peak.

4) There is no conclusive contribution of knowledge, since the results they obtain are very similar to those they had previously, and some even show a low correlation.

We have a Rebuttal to Reviewer 3 regarding this because the low correlation itself can be considered as an improvement. Thus, TTCA measurement became more accurate because of the retarded elution.

Reviewer 4 Report

The manuscript by Ito et al. reports a value-added to their original HPLC-UV method (published in 2011) for the analysis of eumelanin and pheomelanin markers obtained by alkaline hydrogen peroxide oxidation.  

Quantification of both pigments is a matter of interest since their biological roles depends on their content and structure. Authors are experts in the subject and have already contributed different melanins quantification methods since 1985 using HPLC and that using a variety of biological samples. The method presented in this manuscript using ion-pair chromatography is better adapted to complex samples since it presents better resolution of TTCA pheomelanin marker from potential interferences especially in case of low TTCA concentration.

 The title and the abstract adequately correspond to the manuscript. The manuscript is logically structured and well referenced. The work is well described, presents pertinent results and it is of interest to researchers in the field. Nevertheless, I do have some major concerns and minor corrections (listed below).

Some comments to improve the manuscript are as follows:

Major points:

  • TBA+ in RP-HPLC is generally used to sharpen peaks and improve separation efficiency of anionic analytes separated in HPLC using C18 columns. Since melanin’s markers are small anionic compounds depending on pH, the use of a cationic ion-pairing reagent like TBA is appropriate. However, the pH of mobile phase was kept at 2.1, at which PTCA, PDCA and TDCA markers are probably present in a high percentage of non-ionized species (see figure in the pdf file for an example of theoretical pKa of PTCA and TTCA respectively, calculated using chemicalize.org) which do not explain the ion-pair retention efficiency. In the case of TTCA, the marker that is most impacted in retention compare to their previous method, the elution order went from second to the last position (after PTCA). At pH 2, the TTCA ionic distribution shows around 20% of species with at least one ionized acid function that is closer to the ion-pair mechanism. Also for TTCA, a reduction of split peak was obtained when samples were buffer at pH 3 before injection. Considering all this, I wonder whether the pH of the mobile phase deserves more optimization or at least authors should comment on that in the discussion section. Have you tried other mobile phase pH in the preliminary tests?
  •  
  • For the PTeCA analysis using increased methanol concentration:

    How about retention times of TTCA and TDCA under this condition? Are they well separated from PTCA and PDCA?

  • One of the possible inconvenient is that a dedicating column for melanin’s markers application has to be foreseen. The use of an ion pair reagent could not be completely flushed from the column due to strong hydrophobic interactions. This latter point should be considered.

Minor points:

  • In Table 1, number of repetition is missing.
  • The note in Table 2 is incomplete.
  • Line 66 the isomer 3-AHP is mentioned but in figure 1 this one is omitted. Including the chemical structures of 3-AHP in the figure could be helpful to the reader.
  • Line 150: “PDCA, PTeCA, and TTCA.” Should read “PTeCA and TTCA.”

Author Response

First of all, I would like to thank four reviewers for their valuable comments.

1) We added one person (Dr. Tomohisa Hirobe) as co-author because we performed the additional experiment by using the mouse hair samples supplied from him.

2) We added the new Figure 4 showing the correlations between both methods for AHPO of mouse hair, Supplemental Table S2 showing comparison of levels of melanin markers in mouse hair samples between both methods, and Supplemental Figure S3 showing pKa of five markers.

3) Figure 3 was amended.

4) We added the new Table 1 showing validation of the improved method with respect to linearity of calibration, LOD, and LOQ.

Reviewer 4:

The manuscript by Ito et al. reports a value-added to their original HPLC-UV method (published in 2011) for the analysis of eumelanin and pheomelanin markers obtained by alkaline hydrogen peroxide oxidation.  

Quantification of both pigments is a matter of interest since their biological roles depends on their content and structure. Authors are experts in the subject and have already contributed different melanins quantification methods since 1985 using HPLC and that using a variety of biological samples. The method presented in this manuscript using ion-pair chromatography is better adapted to complex samples since it presents better resolution of TTCA pheomelanin marker from potential interferences especially in case of low TTCA concentration.

 The title and the abstract adequately correspond to the manuscript. The manuscript is logically structured and well referenced. The work is well described, presents pertinent results and it is of interest to researchers in the field. Nevertheless, I do have some major concerns and minor corrections (listed below).

Some comments to improve the manuscript are as follows:

Major points:

  • TBA+in RP-HPLC is generally used to sharpen peaks and improve separation efficiency of anionic analytes separated in HPLC using C18 columns. Since melanin’s markers are small anionic compounds depending on pH, the use of a cationic ion-pairing reagent like TBA is appropriate. However, the pH of mobile phase was kept at 2.1, at which PTCA, PDCA and TDCA markers are probably present in a high percentage of non-ionized species (see figure in the pdf file for an example of theoretical pKa of PTCA and TTCA respectively, calculated using chemicalize.org) which do not explain the ion-pair retention efficiency. In the case of TTCA, the marker that is most impacted in retention compare to their previous method, the elution order went from second to the last position (after PTCA). At pH 2, the TTCA ionic distribution shows around 20% of species with at least one ionized acid function that is closer to the ion-pair mechanism. Also for TTCA, a reduction of split peak was obtained when samples were buffer at pH 3 before injection. Considering all this, I wonder whether the pH of the mobile phase deserves more optimization or at least authors should comment on that in the discussion section. Have you tried other mobile phase pH in the preliminary tests?

We would like to acknowledge on these variable comments.

We did not include two papers by Nuta et al. and Matsunaka et al. in the original manuscript because the paper by Nuta et al. was written in Japanese, and the paper by Matsunaka et al. was the application of the paper by Nuta et al. (mobile phase for PTCA analysis: 0.01 M NaH2PO4, 10 mM tetrapropylammonium bromide, pH 2.5/MeOH (90:10), UV 269 nm) and analyzed only PTCA. However, we now realized that those two related papers need to be cited for better understanding of how our method was developed.

We added the following sentences:

Line 84: Nuta et al. and Matsunaka et al. have reported the HPLC conditions using another ion pair reagent, tetra n-propylammonium bromide [18, 19].

Line 184: Nuta et al. [18] examining the effects of ion pair reagents for anionic compounds showed that PTCA peak is retarded several-fold more by TBA+Br- than tetra n-propylammonium bromide and PTCA is retarded more at lower pH. Based on these preliminary results, we chose to add 1 mM TBA+Br- to the original HPLC condition of 0.1 M potassium phosphate buffer, pH 2.1 [7].

Line 205: Using the application software by chemicalize.org (https://chemicalize.com/welcome), the strongest acidic pKa of PDCA, TDCA, PTCA, TTCA, and PTeCA were 3.64, 3.22, 3.20, 2.70, and 1.11, respectively (Figure S3). The microspecies distribution (%) of mono anionic species at pH 2.1, PDCA, TDCA, PTCA, TTCA, and PTeCA, were calculated to be 2.8%, 7.1%, 7.4%, 20.9%, and 91.1%, respectively. This distribution of mono anionic species corresponds exactly to the order of solution in HPLC: PDCA < TDCA < PTCA < TTCA < PTeCA.

2) For the PTeCA analysis using increased methanol concentration:

How about retention times of TTCA and TDCA under this condition? Are they well separated from PTCA and PDCA?

We changed the following sentence:

Line 104: Under those conditions, PTeCA eluted at 31.3 min while PDCA, TDCA, TTCA and PTCA eluted at 6.4, 6.4, 6.8 and 7.2 min, respectively; in addition to PTeCA, the major melanin marker PTCA was well separated from other three markers.

3) One of the possible inconvenient is that a dedicating column for melanin’s markers application has to be foreseen. The use of an ion pair reagent could not be completely flushed from the column due to strong hydrophobic interactions. This latter point should be considered.

What reviewers said about the disadvantages of ion pair HPLC is certainly correct. For example, gradient elution is a difficult task in ion pair chromatography. Slow equilibration between mobile phase and column is another issue and it may take up to some hours for equilibration. Also ion pair reagents due to strong hydrophobic interactions cannot be completely flushed from the column even after extensive washing. This means that dedicating a column to a particular ion pair application is very important. Since ion pair reagents also show significant absorbance in the lower UV region, the development of method in lower UV wavelength region might be a difficult task due to interferences in absorbance from the mobile phase components. In our experiment the use of an ion pair reagent TBA+Br- removes most interfering compounds because the improved HPLC method utilizes 17% (v/v) methanol in place of the 1% (v/v) methanol used in the original HPLC method.

Minor points:

  • In Table 1, number of repetition is missing.

Table 1 was newly added to the text, so old Table 1 becomes Table 2.

We added the number = 5 in Table 2.

  • The note in Table 2 is incomplete.

Table 1 was newly added to the text, so old Table 2 becomes Table 3.

We improved the note in Table 3.

  • Line 66 the isomer 3-AHP is mentioned but in figure 1 this one is omitted. Including the chemical structures of 3-AHP in the figure could be helpful to the reader.

We included newly 4-AHP (+3-AHP) in the figure 1.

  • Line 150: “PDCA, PTeCA, and TTCA.” Should read “PTeCA and TTCA.”

Line 166: We corrected this.

Round 2

Reviewer 1 Report

The authors have revised their manuscript according to the comments.

The abbreviation have to define at the first time, however, the abbreviation is used after that. Authors have to check the manuscript.

Reviewer 3 Report

I consider that the authors have highly improved the manuscript. My concerns were resolved.